# Deeply Fusing Semantics and Interactions for Item Representation Learning via Topology-driven Pre-training

Shiqin Liu
Zhejiang University
College of Computer Science and
Technology
Hangzhou, China
sqliu@zju.edu.cn

Chaozhuo Li*
Beijing University of Posts and
Telecommunications
Key Laboratory of Trustworthy
Distributed Computing and Service
(MoE)
Beijing, China
lichaozhuo@bupt.edu.cn

Xi Zhang
Beijing University of Posts and
Telecommunications
Key Laboratory of Trustworthy
Distributed Computing and Service
(MoE)
Beijing, China
zhangx@bupt.edu.cn

Minjun Zhao
Zhejiang University
College of Computer Science and
Technology
Hangzhou, China
minjunzhao@zju.edu.cn

Yuanbo Xu
Jilin University
College of Computer Science and
Technology
Changchun, China
yuanbox@jlu.edu.cn

Jiajun Bu*
Zhejiang University
College of Computer Science and
Technology
Hangzhou, China
bjj@zju.edu.cn

## Abstract

Learning item representation is crucial for a myriad of on-line e-commerce applications. The nucleus of retail item representation learning is how to properly fuse the semantics within a single item, and the interactions across different items generated by user behaviors (e.g., co-click or co-view). Product semantics depict the intrinsic characteristics of the item, while the interactions describe the relationships between items from the perspective of human perception. Existing approaches either solely rely on a single type of information or loosely couple them together, leading to hindered representations. In this work, we propose a novel model named TESPA to reinforce semantic modeling and interaction modeling mutually. Specifically, collaborative filtering signals in the interaction graph are encoded into the language models through fine-grained topological pre-training, and the interaction graph is further enriched based on semantic similarities. After that, a novel multi-channel co-training paradigm is proposed to deeply fuse the semantics and interactions under a unified framework. In a nutshell, TESPA is capable of enjoying the merits of both sides to facilitate item representation learning. Experimental results of on-line and off-line evaluations demonstrate the superiority of our proposal.

*Correspondence to: Chaozhuo Li <lichaozhuo@bupt.edu.cn>, Jiajun Bu <bjj@zju.edu.cn>.

## CCS Concepts

• **Computing methodologies** → **Artificial intelligence**; • **Information systems** → **Data mining**.

## Keywords

Recommender Systems; Item Representation Learning; Graph Neural Network

**ACM Reference Format:**
Shiqin Liu, Chaozhuo Li, Xi Zhang, Minjun Zhao, Yuanbo Xu, and Jiajun Bu. 2024. Deeply Fusing Semantics and Interactions for Item Representation Learning via Topology-driven Pre-training. In *Proceedings of the 32nd ACM International Conference on Multimedia (MM '24), October 28–November 1, 2024, Melbourne, VIC, Australia.* ACM, New York, NY, USA, 10 pages. https://doi.org/10.1145/3664647.3681639

## 1 Introduction

With the accumulation of an overwhelming amount of information on the Internet, it becomes obligatory for web applications to proactively serve users items based on personalized preferences. To this end, it is crucial to learn high-quality representations for retail items, which are capable of capturing the items' intrinsic semantics precisely. The desirable item representations facilitate a myriad of real-life applications such as retail recommendation [19, 28].

Conventional approaches usually focus on attribute modeling to capture the inner semantics of the items (e.g., title, category, and price) [25, 33, 40, 43, 44] as shown in Figure 1. A common approach is to design delicate and sophisticated representation learning models to encode such attributes into low-dimensional embeddings. In addition to the item attributes, there exists another type of information: user behaviors (e.g., click and view), providing valuable cross-item interactions from human knowledge and preferences as shown in Figure 1. These two types of information mutually reinforce each other to learn quality item representations. For example, as the famous example on Walmart, users tend to co-purchase semantically different items such as beer and diapers,

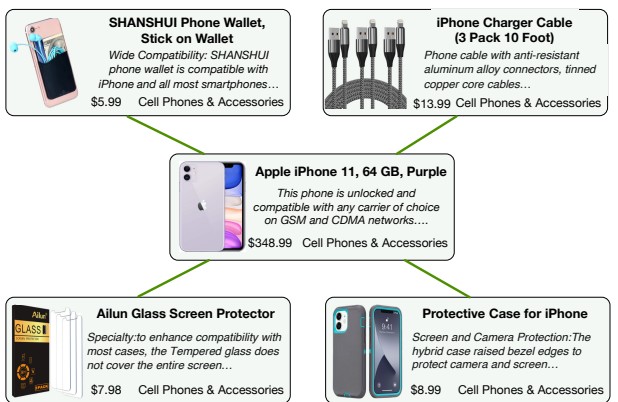

**Figure 1: An illustration of the item attributes and interactions. The green lines denote the co-click relations.**

which reveals that user behaviors provide valuable collaborative filtering information. Meanwhile, if a user likes to eat cherries and interacts with a lot of cherry-related foods, then he/she also prefers the beer with the flavor of cherry based on semantic similarity.

In a nutshell, the core of item representation learning is how to effectively model and fuse the intra-item attributes and inter-item interactions (e.g., co-click or co-view) defined by user behaviors. Traditional works only use the inter-item interactions to define the objective function, which is insufficient to fully uncover the valuable user behaviors [33, 43]. Recently, graph neural networks (GNNs) [17] are introduced to incorporate the rich interactions into the modeling process [7, 48]. Each item is viewed as a node with rich attributes, and the inter-item interactions are viewed as edges. The motivation lies in aggregating the neighborhood as the complementary to enrich the information within a single item. By fusing the item attributes and user behaviors through the message-passing mechanism, GNN-based methods generally achieve superior performance [7]. The fundamental components of the GNN-based methods are the NLU module (e.g., BERT) to model the item semantics and the GNN module to capture the topological interactions. As introduced by the recent works [41], each item can be represented based on its textual feature in the first place and then aggregated with its co-clicked items' embeddings for the final representation. However, it is worth noting that the modeling of textual and topological features are loosely coupled within the above workflow: the item cannot refer to its neighborhood while encoding its own textual feature. Therefore, the fusion of multi-view information might be insufficient.

In addition to the simple combination of GNN and NLU modules, we are going further to investigate how to effectively reinforce each other. On the one hand, from the perspective of NLU models, the pre-trained models are usually learned on the general corpus (e.g., Wikipedia) to capture the common semantic knowledge. However, the semantic similarities between items might be different from common sense. For example, a general pre-trained BERT model would treat the beer and diaper with distinct semantic meanings and learn distant embeddings for them. However, based on the signals from user behaviors, these two items are often co-purchased, and thus their embeddings should be close and similar. Thus, if we could teach language models to understand such interaction signals through continuous pre-training, it has great potential to

boost model performance. On the other hand, the semantics learned by NLU models can also provide complementary information to facilitate GNNs. The graph topology usually suffers from the severe challenge of sparsity as users tend to buy hot and popular items, while the rest long-tail items cannot benefit from the behavior graph. The semantic similarities between the items are capable of building another type of correlation to enhance the cross-item message passing, which contributes to tackling the sparsity.

In this paper, we investigate the problem of item representation learning by mutually reinforcing semantic modeling and interaction modeling. A novel training paradigm TESPA is proposed to reinforce the NLU and GNN modules reciprocally. TESPA is composed of three major components: topology-driven pre-training of language models, semantic-based graph enrichment, and multi-channel co-training module. In the topology-driven pre-training phase, TESPA continues to pre-train the NLU model based on the multi-grained topological signals. Then, the original graph topology is enriched based on semantic similarity. Finally, a novel tightly coupled co-training paradigm is proposed to deeply fuse the semantics and interactions in a multi-channel manner. Contrastive learning techniques are further introduced into both intra-channel and inter-channel contexts to reinforce collaborative correlations. We select the item-to-item (I2I) recommendation as the experimental scenario, and the results on four datasets demonstrate the superior of TESPA. TESPA is further evaluated on a famous on-line advertising platform and achieves significant gains (e.g., +9.25% CTR). Our contributions are summarized as follows:

- We propose a novel mutually reinforced training paradigm for item representation learning, which is capable of enjoying the merits of the item semantics and interactions.
- We design a novel continuous pre-training phase to teach language models to understand the interactions, and a graph enrichment module to alleviate the challenge of topological sparsity based on semantic similarity.
- We conduct extensive experiments over four datasets, and our proposal consistently outperforms SOTA baselines over all the datasets.

## 2 Problem Definition

In this section, we formally define the studied problem. The item dataset is defined as $\mathbf{D} = \{\mathbf{P}, \mathbf{E}\}$, in which $\mathbf{P} = \{p_1, p_2, \ldots, p_K\}$ denotes the set of $K$ items. Each item is associated with a set of attributes such as title, description, price, and category. Matrix $\mathbf{E} \in \mathbb{R}^{K \times K}$ contains the interactions (e.g., co-click) between items. Given a candidate pair $p_i$ and $p_j$, we aim to learn a preference score $\mathbf{f}_{p_i, p_j} \in \{0, 1\}$ to indicate how likely the item $p_j$ should be recommended when user interacted with item $p_i$. Different from existing pure attribute-based or loosely-coupled approaches, here we aim to mutually reinforce the semantic modeling and interaction modeling to achieve quality item representations.

## 3 Methodology

Figure 2 shows the framework of the TESPA model. Given the item attributes and interaction graph, TESPA first continuously pre-trains the general language models to capture the topological signals in the interaction graph. After that, the semantic-based graph enrichment generates extra edges as complementary based on

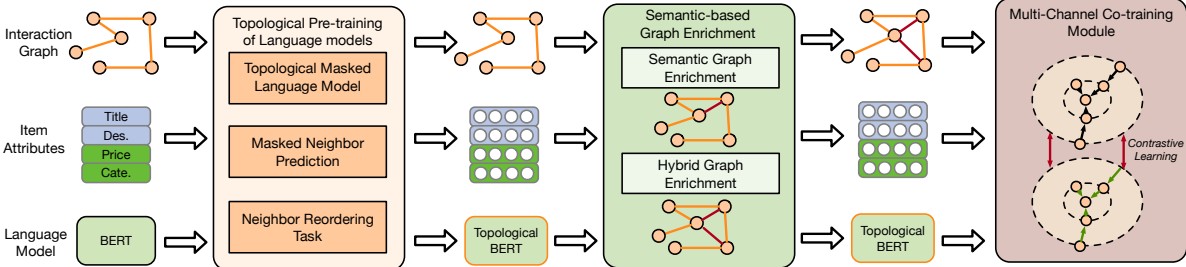

**Figure 2: Framework of the proposed TESPA model.**

the semantic similarities. Finally, the item attributes and interaction graph are co-modeled under a multi-channel training paradigm.

## 3.1 Topology-driven Pre-training of LMs

Traditional pre-trained language models are typically trained on broad text datasets like Wikipedia, aiming to encompass general semantic knowledge. However, as highlighted in the introduction, interaction graphs may encompass distinctive information, exemplified by unique connections like the co-purchase correlations between beer and diapers. Integrating such topology-specific knowledge into language models could enhance item representations. The schematic representation in Figure 3 illustrates our approach, wherein we propose three pre-training tasks to tailor general pre-trained language models to interaction graphs.

**Topological Masked Language Model (TMLM)**: Previous work [5] demonstrates that the continuous pre-training to incorporate domain-specific knowledge is capable of boosting the expressiveness of language models. Here we modify the original Masked Language Model (MLM) [2] by incorporating the topological connections, dubbed Topological Masked Language Model (TMLM). Given a center item $c$ and one of its neighbors $n$, their descriptive text is formally defined as $\mathbf{T}^{(\mathbf{c})} = \{t_1^{(c)}, t_2^{(c)}, \ldots, t_k^{(c)}\}$ and $\mathbf{T}^{(\mathbf{n})} = \{t_1^{(n)}, t_2^{(n)}, \ldots, t_u^{(n)}\}$, respectively. We randomly replace a subset of tokens in the center item $\mathbf{T_c}$ with a special token [MASK]. The objective of TMLM is to predict the masked tokens. Let $\Phi^{(c)} = \{\phi_1^{(c)}, \phi_2^{(c)}, ..., \phi_m^{(c)}\}$ represent the indexes of the $m$ masked tokens in the sentence $\mathbf{T}^{(c)}$. Let $\mathbf{T}_\Phi^{(c)}$ denotes the set of masked tokens in $\mathbf{T}^{(c)}$, and $\mathbf{T}_{-\Phi}^{(c)}$ denotes the set of observed (unmasked) tokens. The objective of TMLM is formally defined as:

$$\mathcal{L}_{\text{tmlm}}(\mathbf{T}_\Phi^{(c)}|\mathbf{T}_{-\Phi}^{(c)}, \mathbf{T}^{(n)}) = \frac{1}{m}\sum_{i=1}^{m}\log p(t_{\phi_i}|\mathbf{T}_{-\Phi}^{(c)}, \mathbf{T}^{(n)}; \theta), \quad (1)$$

in which $\theta$ denotes the learnable parameters in the pre-trained models. TMLM empowers the language model with the capability of predicting the missing tokens in the neighborhoods, which smoothly encodes the token correlations across different items.

**Masked Neighbor Prediction (MNP)**: The previous TMLM task focuses on the token-level correlations. Here we propose a Masked Neighbor Prediction (MNP) task to encode the topological signals at the item level. MNP aims to predict whether the two items will be co-clicked. MNP is similar to the Next Sentence Prediction (NSP) task [2], but in MNP, part of the item's textual descriptions are randomly masked, which increases the prediction difficulty due to the corrupted semantics. Therefore, it forces the model to seek complementary information from the co-clicked items, which will

further strengthen the model's capability of leveraging the neighborhood. Given a center item $c$ and one of its co-clicked neighbors $n$, we first randomly mask a portion of tokens from the original text $\mathbf{T}^{(c)}$ and $\mathbf{T}^{(n)}$. Then, their representations are defined as the [CLS] embeddings learned by the language models as $\hat{\mathbf{h}}_c$ and $\hat{\mathbf{h}}_n$. We adopt the link prediction as the training task, and the objective function is formally defined as follows: $\mathcal{L}_{mnp} = -\log\frac{e^{\langle\hat{h}_n,\hat{h}_c\rangle}}{e^{\langle\hat{h}_n,\hat{h}_c\rangle}+\sum_{r\in R}e^{\langle\hat{h}_n,\hat{h}_r\rangle}}$, in which $\langle\cdot\rangle$ denotes the inner-item operator. Set $R$ contains the negative samples. Conventional randomly sampled negative samples might be easily filtered and further hinder the discriminative capability. Here we design a hard negative sampling method based on the topology distances. Given a center item, we view its neighbors within a fixed distance window (e.g., 3) but are not directly connected as the hard negative samples.

**Neighbor Reordering Task (NRT)**: MNP task captures the first-order relations between items. However, one of the striking characteristics of the graph is its nature of high-order connections. The high-order relations provide the relative positions of the neighboring nodes. Hence, we designed another task, dubbed neighbor Reordering task, to capture the high-order closeness. Given a center item $c$, we first randomly select a set of neighbors from its topological context based on the shortest path distance (SPD). Specifically, a single neighbor $n^{(s)}$ is randomly selected from the $s$-hop neighbors with $\text{SPD}(c, n^{(s)}) = s$. After collecting neighbors from all the $u$ hops, we can achieve a set of neighbors from different hops: $S = \{n^{(1)}, n^{(2)}, \ldots, n^{(u)}\}$, where $u$ is a hyper-parameter deciding the range of receptive field. Based on the center node $c$ and the neighbor set $S$, we formalize the NRT task as a classification task to classify each pair $< c, n^{(s)} >$ to the correct category $s$ denoting their relative distance. The representations of $c$ and $n^{(s)}$ are concatenated together and fed into a linear classifier. The training objective is defined as the cross-entropy loss:

$$\mathbf{e} = \sigma(\mathbf{W}_{nrt} \times \text{concat}(\mathbf{h}_c, \mathbf{h}_s)),$$
$$\mathcal{L}_{nrt} = -\sum_{i=1}^{u}\mathbf{y}_i\log(\mathbf{e}_i), \quad (2)$$

where $\mathbf{W}_{nrt}$ is the weight matrix to project the embeddings to the $u$-dimensional vectors. $\sigma$ denotes the softmax activation function. $\mathbf{y}$ is the ground truth label. NRT task enables the pre-trained model to learn relative high-order relationships among the neighbors.

**Multi-task Training Paradigm**: In order to incorporate topological information into language models, we introduce three pre-training tasks: TMLM, MNP, and NRT. Each task contributes to capturing distinct types of topological information. A unified multi-task training paradigm becomes essential to harness the benefits

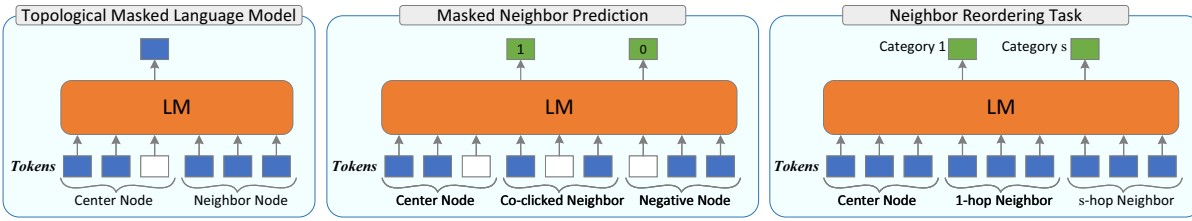

**Figure 3: Topology-driven pre-training of LMs.**

offered by these diverse tasks. While a straightforward strategy involves training these tasks sequentially, this approach may encounter the challenge of Knowledge Forgetting, as highlighted in previous research [22]. In such a scenario, the learned model tends to prioritize the final task, potentially forgetting knowledge acquired from preceding tasks. To address this issue, we propose a multi-task training strategy that iteratively trains batches associated with different tasks. This involves offline generation of training samples for each task, with the inclusion of a special token to identify task affiliation. Subsequently, batches corresponding to different tasks are sequentially presented to the model during the training process. The classification layer and objective function are adaptively selected based on the input task identity. Essentially, the proposed strategy ensures the equitable training of the three tasks at the batch level, mitigating the risk of knowledge forgetting.

## 3.2 Semantic-based Graph Enrichment

While the preceding section delves into the incorporation of topological information into language models, our focus extends to enhancing graph topology through semantic similarities. User behavior, characterized by a preference for trending and popular items, results in a power-law distribution of node degrees [45]. This distribution implies that a majority of items have limited connections (e.g., on a prominent advertising platform, over 64% of online items receive fewer than three clicks in a month). Consequently, these tail nodes do not fully benefit from topological connections. To address this limitation, our objective is to augment the original graph topology by introducing new edges based on semantic similarities. Metric learning is employed to generate the semantic graph:

$$\mathbf{E_s}[i, j] = \begin{cases} \mathcal{F}(\mathbf{h_i}, \mathbf{h_j}) & \mathcal{F}(\boldsymbol{h_i}, \boldsymbol{h_j}) \geq \epsilon, \\ 0 & \mathcal{F}(\boldsymbol{h_i}, \boldsymbol{h_j}) < \epsilon, \end{cases} \quad (3)$$

where $\mathbf{E_s} \in \mathbb{R}^{K \times K}$ is the learned semantic graph, and $\mathbf{h_i}$ denotes the embedding of item $p_i$ learned by the pre-trained models. $\epsilon$ is a hyperparameter to control the sparsity of the learned semantic graph. The function $\mathcal{F}$ is the similarity calculation function. To ensure the model efficiency, here $\mathcal{F}$ is defined as the cosine similarity, and thus the nearest neighbor search could be solved by the efficient ANN search algorithms [35].

In addition, based on the motivation that items with similar semantics tend to share similar neighbors, the hybrid graph is further proposed to generate new edges by deeply fusing the semantics and graph connections. Based on the original graph $\mathbf{E}$ and the semantic graph $\mathbf{E}_s$, the hybrid graph $\mathbf{E}_h$ is formulated as: $\mathbf{E_h} = \mathbf{E}_s\mathbf{E}$. The semantic closeness is propagated through the topology connections to provide rich hybrid correlations.

Overall, we can achieve three types of graphs: the original graph $\mathbf{E}$, the semantic graph $\mathbf{E}_s$ and the hybrid graph $\mathbf{E}_h$, which are combined together as the final enriched topological structure.

## 3.3 Multi-channel Co-training Module

Based on the pre-trained language models and the enriched interaction graphs, we further introduce the multi-channel co-training module, as shown in Figure 4, to aggregate the item attributes and the interaction graph. Traditional GNNs [4, 13, 37] usually encode the node attributes into static embeddings, which are frozen in the GNN training process. Such pre-learned static embeddings cannot be ensured to align with the downstream tasks, which may hinder the recommendation performance. Thus, we propose to co-train the item attribute modeling module and the topology aggregation module. The major challenge lies in the heterogeneity of the interaction graph. First, the item has a set of attributes such as title, description, category, and price. These attributes have distinct data formats and various characteristics. Conventional GNN models usually encode all the attributes into a single item embedding, and aggregate such representations from the neighborhood. However, the amount of information in different attributes might be different. For example, compared to the long title and description, the information within the price and category has a larger chance of being overwhelmed in the aggregation process. Thus, we propose a multi-channel aggregation paradigm to preserve the fine-grained correlations. Namely, each type of item attribute is viewed as an independent channel to conduct modeling and aggregation. The outputs from the final layer of different channels are combined as the final representation. For the sake of clarity, we use the semantic channel to describe the channels of title and description, and the categorical channel contains the price and category.

### 3.3.1 Semantic Channel.
In the semantic channel, we aim to co-learn the NLU and GNN models in a deeply coupled manner. Similar to [20], we adopt the tightly-coupled learning paradigm in which multiple layers of graph encoder and textual encoder are alternately stacked. A [CLS] token is padded in the front of each text as the sentence representation.

For the graph encoder, the input interaction graph contains three types of edges generated by the graph topology enrichment. Thus, we design the heterogeneous graph aggregations to capture these fine-grained correlations. In the $l$-th layer of graph encoder, the [CLS] token embeddings of the center node and its neighbors are collected as a matrix $\mathbf{M}^{(l-1)}$. The homogeneous graph encoder would be formalized as follows:

$$\mathbf{A}^{(l-1)} = \frac{\mathbf{Q}^{(l-1)}\mathbf{K}^{(l-1)\top}}{\sqrt{d}},$$
$$\hat{\mathbf{M}}^{(l-1)} = \text{softmax}\left(\mathbf{A}^{(l-1)}\right)\mathbf{V}^{(l-1)}, \quad (4)$$

where

$$\begin{cases} \mathbf{Q}^{(l-1)} & = \mathbf{M}^{(l-1)}\mathbf{W}_Q^{(l-1)}, \\ \mathbf{K}^{(l-1)} & = \mathbf{M}^{(l-1)}\mathbf{W}_K^{(l-1)}, \\ \mathbf{V}^{(l-1)} & = \mathbf{M}^{(l-1)}\mathbf{W}_V^{(l-1)}, \end{cases} \quad (5)$$

in which matrices $\mathbf{W}_Q^{(l-1)}, \mathbf{W}_K^{(l-1)}, \mathbf{W}_V^{(l-1)} \in \mathbb{R}^{d \times d}$ denote the trainable variables. Here we further modify its architecture to fit the heterogeneous setting. Assume item $p_i$ and $p_j$ is connected by the semantic edge in $\mathbf{E}_s$ with the weight $c_{ij}$. The original attention is modified as follows:

$$\mathbf{A}^{(l-1)}[i, j] = \frac{(\mathbf{h}_i^{(l-1)}\mathbf{W}_{Qs}^{(l-1)})(\mathbf{h}_j^{(l-1)}\mathbf{W}_{Ks}^{(l-1)})^\top}{\sqrt{d}} + w_s c_{ij}, \quad (6)$$

in which $s$ denotes the type of connected edge. We utilize different attention parameters for different types of edges, which contributes to capturing the heterogeneity. In addition, the input graph is essentially a weighted graph, and the weights of edges denote the degrees of closeness. Thus, the graph encoder incorporates the weights via a bias term to the attention module. In the graph encoder, the center node and the neighbor nodes can reciprocally exchange messages, which embed the topological interaction information into the learned [CLS] embeddings.

Then in the textual encoder, the topological [CLS] embeddings are dispatched to the related nodes. For each node, we can achieve a matrix $\mathbf{M}_s^{(l-1)} \in \mathbb{R}^{(m+1) \times d}$ by consolidating the [CLS] embedding and the embeddings of $m$ original tokens. Matrix $\mathbf{M}_s^{(l-1)}$ would be fed into the textual encoder similar to Equation (4) to convey the neighborhood information from the [CLS] token to the original textual tokens. By these alternately stacked textual and topological encoders, the semantics within a single item and the interactions are deeply fused to learn powerful item representations.

*3.3.2 Categorical Channel.* For the categories and prices of the items, we view them as categorical features. The embeddings of the categories are randomly initialized as the look-up table and will be updated to fit the training signals. For the prices, we segment and sort these float values into bins. The identities of the belonged bins are viewed as categorical features, and thus the continuous features are converted to categorical ones. The topological aggregation process of the categorical channels is the same as Equation (6) by incorporating the types and weights of the edges.

*3.3.3 Multi-Channel Aggregation with Contrastive Learning.* We can achieve an embedding vector for each channel that preserves semantics and interactions. Considering that the informativeness of various channels might be different, the attention mechanism is further employed to learn the distinct importance of channels. Assume $\mathbf{h}_p^t, \mathbf{h}_p^d, \mathbf{h}_p^p$ and $\mathbf{h}_p^c$ denote the representations learned from the title channel, description channel, price channel and category channel, respectively. For instance, the attention score of the title channel is defined as follows:

$$a_t = \sigma(\mathbf{w_w} \cdot \mathbf{h}_p^t + b_w),$$
$$\alpha_t = \frac{\exp(a_t)}{\exp(a_t) + \exp(a_d) + \exp(a_c) + \exp(a_p)}, \quad (7)$$

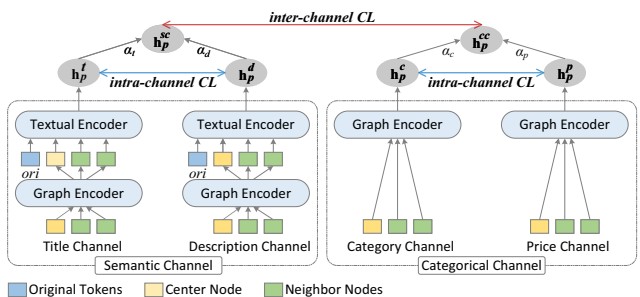

**Figure 4: Multi-channel co-training module.**

in which $\mathbf{w_w} \in \mathbb{R}^d$ and $b \in \mathbb{R}$ are the parameters in the attention network. $\sigma$ is the activation function. $\alpha_t$ represents the normalized importance of the title channel. The final representation of the item is the weighted sum of the representations of different channels based on the learned attention weights:

$$\mathbf{h} = \alpha_t \mathbf{h}_p^t + \alpha_d \mathbf{h}_p^d + \alpha_c \mathbf{h}_p^c + \alpha_p \mathbf{h}_p^p. \quad (8)$$

The enhanced efficacy of the multi-channel mechanism arises from the presence of distinct types of information within each channel. This allows the model to discern the intrinsic characteristics of individual channels and glean valuable insights. Nevertheless, the intricate relationships and collaborative correlations among different channels have been underexplored. To address this gap, we introduce contrastive learning techniques into both intra-channel and inter-channel contexts to reinforce collaborative correlations between various channels.

Contrastive learning is systematically incorporated both intra- and inter-channel, specifically within the semantic and categorical channels. The objective is to reinforce collaborative associations within each channel and foster connectivity between the two channels. To capture the collaborative information within each channel, taking the semantic channel as an example, we design an intra-channel contrastive loss with contrastive instance $(\mathbf{h}_p^t, \mathbf{h}_p^d)$. The corresponding contrastive learning loss is formally defined as:

$$\mathcal{L}_{intra}^S = \sum_{p \in \mathbf{P}} -\log \frac{exp(s(\mathbf{h}_p^t, \mathbf{h}_p^d)/\tau)}{\sum_{p_{neg} \in \mathbf{P}} exp(s(\mathbf{h}_p^t, \mathbf{h}_{p_{neg}}^d)/\tau)}, \quad (9)$$

where $\tau$ denotes the temperature in softmax. $s(\cdot)$ denotes a cosine similarity function that quantifies the similarity between two representations. $\mathcal{L}_{intra}^S$ denotes the contrastive loss within the semantic channel. Turning attention to the categorical channel, contrastive instances are obtained from both the category and price channels. The analogous contrastive learning loss for the categorical channel, denoted as $\mathcal{L}_{intra}^C$, is derived in a similar manner.

In addition, to capture the correlations between the semantic channel and the categorical, we further design an inter-channel contrastive learning loss. Contrastive instances between the semantic channel and the categorical channel are obtained by aggregating the corresponding embeddings:

$$\mathbf{h}_p^{sc} = \alpha_t \mathbf{h}_p^t + \alpha_d \mathbf{h}_p^d, \quad (10)$$
$$\mathbf{h}_p^{cc} = \alpha_c \mathbf{h}_p^c + \alpha_p \mathbf{h}_p^p, \quad (11)$$

where $\alpha$ represents the normalized importance of the corresponding channel. Subsequently, the obtained embeddings $\mathbf{h}_p^{sc}$ and $\mathbf{h}_p^{cc}$ serve

**Table 1: Statistics of four datasets.**

| Dataset | #item | #training | #validation | #test |
|---|---|---|---|---|
| ProductAds | 2,274,778 | 18,478,012 | 50,000 | 262,841 |
| ProductReco | 10,745,783 | 103,295,927 | 50,000 | 942,426 |
| Amazon-sports | 28,316 | 456,050 | 6,745 | 1,540 |
| Yelp | 2,639 | 185,097 | 2,731 | 5,492 |

as contrast instances, and the corresponding contrastive learning loss is defined as follows:

$$\mathcal{L}_{inter} = \sum_{p \in \mathbf{P}} -log \frac{exp(s(\mathbf{h}_p^{sc}, \mathbf{h}_p^{cc})/\tau)}{\sum_{p_{neg} \in \mathbf{P}} exp(s(\mathbf{h}_p^{sc}, \mathbf{h}_{p_{neg}}^{cc})/\tau)}, \quad (12)$$

where $\mathcal{L}_{inter}$ denotes the contrastive loss across the semantic and categorical channels.

*3.3.4 Training Objective Function.* The BPR loss [29] is selected as the training objective function, which measures the relative order of the positive pairs and negative pairs. BPR assumes that the recommendation score of the positive node pairs should be higher than the corresponding negative ones. The objective function $\mathcal{L}_{\mathcal{BPR}}$ of our model is formulated as:

$$\mathcal{L}_{\mathcal{BPR}} = \sum_{(p,p^+,p^-) \in O} -\ln \sigma(s_{(p,p^+)} - s_{(p,p^-)}), \quad (13)$$

where $O = \bigcup_m \{(p, p^+, p^-)|(p, p^+) \in \mathbf{E}, (p, p^-) \in \mathbf{E}^-\}$ denotes the training samples. In our implementation, we take advantage of "in-batch negative samples" [16, 24] to reduce the computation cost.

# 4 Experiments

## 4.1 Datasets

We use four datasets to evaluate the performance of the proposed TESPA model. Datasets of ProductAds and ProductReco are collected from different niche markets of a popular on-line advertising platform. We first collect the users' click behaviors within a fixed time interval (e.g., six months) and then construct the co-click graph. Each node in the co-click graph denotes an item, while the weights on the edges indicate the co-click frequency. To reduce the influence of user behaviors' uncertainty, we remove the edges with weights less than 3. After that, the edge weights are normalized into (0,1). Each item is associated with four types of attributes: title, description, category, and price. To split the training and test sets, we randomly select 10% edges from the co-clicked graph as the test set, and the rest edges are viewed as the training set. A fixed number of edges are randomly selected as the validation set. In addition, two popular datasets, Yelp[1] and Amazon-sports[2] are also incorporated. Table 1 presents the detailed data statistics.

## 4.2 Baseline Methods

To extensively evaluate the performance of our proposal, we select four types of methods as the baselines. Firstly, we select three NLU models to conduct item recommendation based on the pure semantic information: DSSM [12], LSTM [14] and BERT [2]. Besides, four popular recommendation models MF [11], NCF [10], NGCF [36] and LightGCN [7] are selected as the baselines. In addition, we also adopt a set of popular GNN models (GCN [17], GAT [32],

GraphSAGE [6]) to verify the importance of the co-training paradigm. Finally, recently several works have been proposed to jointly train the GNNs and NLUs, such as TextGNN [50], AdsGNN [20], HashCODE [27] and Heterformer [15], which are also introduced as the baselines.

## 4.3 Experimental Settings

The evaluation task is defined as a retrieval problem, i.e., to predict whether two input items would be co-clicked based on the historical interactions and their intrinsic attributes. We adopt three typical ranking metrics, including **Precision@1**, **NDCG**, and **MRR**. For each testing case, an item will be associated with 300 candidates: 1 positive plus 299 randomly sampled negatives for evaluation. The fundamental checkpoint of the pre-trained model in all the methods is set to the "Bert-base-uncased" in the huggingface[3]. For the semantic-based models, the prices and categories are viewed as learnable embedding in the lookup tables, and directly concatenated with the outputs from the semantic models. For the GNNs, we use the pre-trained language models to encode the textual information into the low-dimensional attribute vectors. Other item attributes are incorporated into the baseline models in a straight-forward manner, such as concatenation. For the proposed TESPA model, the training batch size is set to 32, the learning rate is set to $1e-5$, the number of training epochs is 3, and the number of layers in graph transformers is set to 3. For the GNN baselines, the number of GNN layers is set to 3. Other hyper-parameters are carefully tuned on the validation dataset. The training is on 16× Nvidia V100-16GB GPUs. We use PyTorch 1.6.0 for implementation.

## 4.4 Off-line Experimental Results

Table 2 presents the off-line performance of different models. We conduct each method five times and report the average performance. From the results, we can easily achieve the following observations: (1) Traditional pure semantic models (LSTM and DSSM) achieve inferior performance compared to other methods, which reveals that user behaviors are crucial to learning desirable item representations. (2) Heterformer and HashCODE are two recent works to co-train the GNN and NLU models under a unified learning framework. Their performance significantly outperforms their counterparts, verifying the effectiveness of the co-training paradigm. (3) Our proposal consistently beats baselines by a large margin over all the metrics. Compared to the best baseline, TESPA improves the performance by 3% on average, which passes the significance testing. The performance improvements are owed to the mutual reinforcement between interactions and item semantics and the appropriate multi-channel graph aggregations.

## 4.5 Ablation Study

*4.5.1 Topological Pre-training Tasks.* In section 3.1, three pre-training tasks are proposed to encode the topological connections into the language models, including the TMLM, MNP and NRT. Here we aim to investigate the importance of different tasks. The traditional MLM and NSP tasks are also adopted as baselines. The NSP task is modified to predict the neighbors as the next sentences. Please note that, the graph enrichment and the co-training module are

---

[1]https://www.yelp.com/dataset
[2]https://nijianmo.github.io/amazon/#subsets

[3]https://github.com/huggingface/transformers/

**Table 2: The off-line performance of the Item2Item recommendations. The improvements of TESPA compared to SOTA baselines are statistically significant (sign test, p-value < 0.01).**

| Model | ProductAds P@1 | NDCG | MRR | ProductReco P@1 | NDCG | MRR | Amazon-sports P@1 | NDCG | MRR | Yelp P@1 | NDCG | MRR |
|---|---|---|---|---|---|---|---|---|---|---|---|---|
| LSTM | 0.376 | 0.578 | 0.532 | 0.348 | 0.521 | 0.495 | 0.103 | 0.253 | 0.242 | 0.114 | 0.201 | 0.242 |
| DSSM | 0.408 | 0.592 | 0.571 | 0.381 | 0.553 | 0.528 | 0.116 | 0.271 | 0.255 | 0.120 | 0.217 | 0.258 |
| BERT | 0.673 | 0.772 | 0.722 | 0.616 | 0.731 | 0.695 | 0.204 | 0.372 | 0.395 | 0.210 | 0.398 | 0.383 |
| GCN | 0.634 | 0.771 | 0.719 | 0.619 | 0.727 | 0.703 | 0.218 | 0.384 | 0.382 | 0.225 | 0.408 | 0.399 |
| GraphSAGE | 0.661 | 0.803 | 0.754 | 0.625 | 0.762 | 0.719 | 0.225 | 0.387 | 0.409 | 0.219 | 0.388 | 0.378 |
| GAT | 0.674 | 0.804 | 0.748 | 0.632 | 0.756 | 0.730 | 0.221 | 0.385 | 0.381 | 0.234 | 0.411 | 0.407 |
| MF | 0.628 | 0.753 | 0.706 | 0.603 | 0.706 | 0.687 | 0.193 | 0.264 | 0.277 | 0.181 | 0.368 | 0.271 |
| NCF | 0.647 | 0.787 | 0.715 | 0.639 | 0.742 | 0.709 | 0.232 | 0.402 | 0.415 | 0.247 | 0.423 | 0.419 |
| NGCF | 0.672 | 0.793 | 0.738 | 0.649 | 0.760 | 0.728 | 0.248 | 0.419 | 0.411 | 0.261 | 0.437 | 0.429 |
| LightGCN | 0.702 | 0.816 | 0.777 | 0.661 | 0.785 | 0.735 | 0.249 | 0.431 | 0.429 | 0.269 | 0.441 | 0.437 |
| TextGNN | 0.737 | 0.836 | 0.782 | 0.682 | 0.791 | 0.757 | 0.263 | 0.477 | 0.464 | 0.274 | 0.478 | 0.464 |
| AdsGNN$_t$ | 0.751 | 0.859 | 0.806 | 0.703 | 0.815 | 0.774 | 0.279 | 0.492 | 0.488 | 0.287 | 0.490 | 0.479 |
| Heterformer | 0.759 | 0.877 | 0.821 | 0.709 | 0.816 | 0.773 | 0.273 | 0.501 | 0.484 | 0.287 | 0.497 | 0.475 |
| HashCODE | 0.763 | 0.864 | 0.819 | 0.718 | 0.822 | 0.789 | 0.285 | 0.510 | 0.496 | 0.299 | 0.504 | 0.488 |
| TESPA | **0.781** | **0.874** | **0.825** | **0.737** | **0.845** | **0.819** | **0.301** | **0.521** | **0.514** | **0.315** | **0.528** | **0.512** |

**Table 3: Ablation study on the continuous pre-training tasks. We remove the graph enrichment and the co-training module in this study to ensure the fairness.**

| Model | ProductAds P@1 | NDCG | MRR | ProductReco P@1 | NDCG | MRR |
|---|---|---|---|---|---|---|
| BERT | 0.643 | 0.772 | 0.722 | 0.616 | 0.731 | 0.695 |
| MLM | 0.655 | 0.779 | 0.733 | 0.623 | 0.739 | 0.701 |
| NP | 0.660 | 0.784 | 0.739 | 0.632 | 0.742 | 0.709 |
| TMLM | 0.662 | 0.787 | 0.745 | 0.646 | 0.754 | 0.718 |
| MNP | 0.668 | 0.797 | 0.758 | 0.659 | 0.761 | 0.727 |
| NRT | 0.675 | 0.816 | 0.768 | 0.665 | 0.769 | 0.733 |
| TMLM+MNP | 0.679 | 0.819 | 0.766 | 0.668 | 0.772 | 0.744 |
| TMLM+NRT | 0.685 | 0.827 | 0.783 | 0.677 | 0.779 | 0.752 |
| MNP+NRT | 0.694 | 0.822 | 0.781 | 0.675 | 0.782 | 0.746 |
| TMLM+MNP+NRT | **0.702** | **0.826** | **0.784** | **0.679** | **0.785** | **0.749** |

**Table 4: Ablation study on the multi-task training paradigm in the topology-driven pre-training phase.**

| Paradigm | ProductAds P@1 | NDCG | MRR | ProductReco P@1 | NDCG | MRR |
|---|---|---|---|---|---|---|
| Task-level | 0.772 | 0.861 | 0.814 | 0.723 | 0.835 | 0.802 |
| Batch-level | **0.781** | **0.874** | **0.825** | **0.737** | **0.845** | **0.819** |

**Table 5: Ablation study on the graph enrichment module.**

| Graph type | ProductAds P@1 | NDCG | MRR | ProductReco P@1 | NDCG | MRR |
|---|---|---|---|---|---|---|
| $\mathbf{E}$ | 0.765 | 0.852 | 0.811 | 0.712 | 0.826 | 0.801 |
| $\mathbf{E} + \mathbf{E}_s$ | 0.772 | 0.862 | 0.814 | 0.725 | 0.838 | 0.809 |
| $\mathbf{E} + \mathbf{E}_h$ | 0.774 | 0.867 | 0.816 | 0.727 | 0.840 | 0.812 |
| $\mathbf{E} + \mathbf{E}_s + \mathbf{E}_h$ | **0.781** | **0.874** | **0.825** | **0.737** | **0.845** | **0.819** |

pre-training tasks capture valuable and unique information from different perspectives.

*4.5.2 Multi-task Training Paradigm.* In section 3.1, we design the batch-level iterative training paradigm. Here the traditional task-level training strategy is introduced as the baseline to verify the effectiveness of our proposal. The task-level strategy successively fully trains the pre-training tasks. Table 4 reports the experimental results. One can see that the performance of the task-level ablation model is degraded compared to the batch-level method, which might because the batch-level iterative training paradigm alleviates the effect of knowledge forgetting.

*4.5.3 Graph Enrichment.* In section 3.2, the original graph topology is strengthened based on the semantic similarity between items, and two types of graphs (i.e., semantic graph $\mathbf{E}_s$ and hybrid graph $\mathbf{E}_h$) are learned as the complementary to the original graph $\mathbf{E}$. Here we study the influence of different types of graphs on the model performance. Experimental results are shown in Table 5. One can easily achieve the following conclusions: (1) Model performance consistently increases with the enriched graphs on all the datasets over all the metrics. This phenomenon reveals that semantic-based relations are capable of providing unique helpful information. (2) The hybrid graph $\mathbf{E}_h$ is slightly better than the semantic graph $\mathbf{E}_s$, which verifies our motivation that the profound fusion of behavior graph and item attributes would advance model performance. (3) TESPA achieves the best performance with all types of graphs,

removed to ensure fairness. Table 3 presents the performance of BERT with different pre-training tasks. The following conclusions could be easily obtained: (1) Topology-aware pre-training tasks (e.g., NP, TMLM and MNP) consistently outperform the pure semantic-based task (e.g., MLM), which verifies the importance of interaction graphs in item representation learning. (2) TMLM and MNP consistently beat their counterparts MLM and NP. The performance gains are reasonable because TMLP can capture the topology information while MNP advances the model expressiveness via the corrupted inputs. (3) Model performance is further improved after combining all the pre-training tasks, which reveals that different

                                          

**Table 6: Ablation study on the multi-channel module.**

| Graph type | ProductAds | | | ProductReco | | |
|---|---|---|---|---|---|---|
| | P@1 | NDCG | MRR | P@1 | NDCG | MRR |
| w/o title | 0.751 | 0.842 | 0.796 | 0.694 | 0.803 | 0.772 |
| w/o description | 0.767 | 0.853 | 0.804 | 0.713 | 0.825 | 0.794 |
| w/o price | 0.769 | 0.858 | 0.809 | 0.719 | 0.827 | 0.804 |
| w/o category | 0.769 | 0.857 | 0.806 | 0.717 | 0.829 | 0.802 |
| w/o intra-channel CL | 0.763 | 0.860 | 0.812 | 0.703 | 0.818 | 0.797 |
| w/o inter-channel CL | 0.769 | 0.858 | 0.809 | 0.702 | 0.827 | 0.795 |
| Single channel Agg. | 0.770 | 0.862 | 0.811 | 0.719 | 0.832 | 0.804 |
| TESPA | **0.781** | **0.874** | **0.825** | **0.737** | **0.845** | **0.819** |

which demonstrates that each graph preserves its unique knowledge and is indispensable to the desired item presentations.

*4.5.4 Multi-channel Co-training Module.* The ablation study is also conducted on the multi-channel co-training module. Firstly, we investigate the importance of different channels. Each channel is removed from the model successively, and the results of different ablation models are reported in Table 6. (1) The title is more important than other attributes. Model performance drops by nearly 3% after removing the title. To attract the user's attention, item titles usually consist of brief but accurate descriptions, and the essential information to highlight the items. (2) The effect of price and category is smaller than the semantic features but still contributes to the unique performance gains. The performance decreases without any single channel, demonstrating the importance of a comprehensive presentation learning model.

Besides, we also investigate the effectiveness of the multi-channel aggregation function. The single-channel aggregation is selected as the baseline, which first encodes all the attributes into an embedding and then propagates this embedding to the neighborhood. As shown in Table 6, the multi-channel method surpasses the single channel model by nearly 1.5% on average. For the single channel approach, attributes with rich semantics might overwhelm the comparatively uninformative ones, leading to information loss and inferior performance.

To assess the efficacy of the contrastive learning components within both inter-channel and intra-channel contexts, we conducted experiments by individually removing these two losses. The resultant model performances are presented in Table 6. Notably, a conspicuous decline in model performance is observed after the removal of these losses. This decline underscores the pivotal role played by the contrastive learning losses in enhancing the model's capability to capture nuanced relationships within and between channels. This deterioration in performance suggests that the effectiveness of the model is intricately tied to the integration of contrastive learning mechanisms. The removal of these losses appears to compromise the model's ability to leverage the inherent relationships and collaborative correlations among different channels, underscoring the importance of the proposed contrastive learning framework in our approach.

## 4.6 Parameter Sensitivity Analysis

Here we study the performance sensitivity of our proposal on two core parameters: the sparsity weight $\epsilon$ in Equation (3) and the training batch size $b$ of Equation (13). As the performance trends on

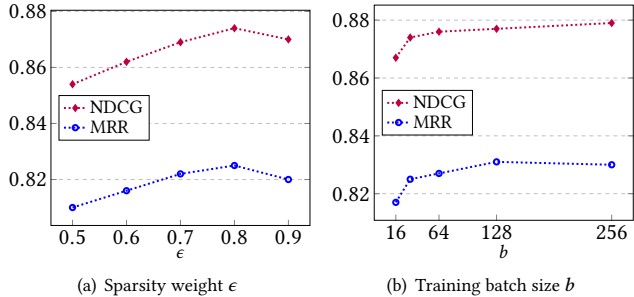

(a) Sparsity weight $\epsilon$       (b) Training batch size $b$

**Figure 5: Parameter sensitivity studies on core parameters.**

the two datasets of ProductAds and ProductReco are similar, here we only report the results on ProductAds dataset. The NDCG and MRR scores under different settings on two datasets are recorded. Figure 5 presents the experimental results. From the left sub-figure, one can see that with the increase of $\epsilon$, the performance over all the datasets first increases and then decreases. A larger $\epsilon$ leads to a sparser graph with limited topology information, while noise may be introduced if $\epsilon$ is too small. Thus, $\epsilon$ should be carefully tuned to achieve desirable performance. From the right sub-figure, we can see that with the increase of $b$, model performance first significantly increases and then slightly increases or keeps steady. This is reasonable as at the beginning, a larger batch size provides more in-batch negative samples to boost the discriminative capability. However, model performance would keep steady when the information in the negative instances is sufficient to provide accurate training signals. The on-line performance analysis can be found in Appendix A, and the case study of online services can be found in Appendix B.

## 5 Related Work

The item representation is one of the crucial issues for intelligent web services. The historical works can be divided into three major categories based on their utilized information: topological-based methods [1, 3, 8, 9, 18, 30, 31, 34, 38, 39, 46], textual-based methods [21, 25, 33, 42–44], and combined-based methods [15, 23, 26, 27, 41, 47, 49, 50]. Topological-based methods rely on topological features, and textual-based methods rely on the textual feature for item representation. Combined-based methods leverage the combination of topological and textual features for better representation quality. Further details on related work can be found in Appendix C.

## 6 Conclusion

In this paper, we extensively study the critical task of item representation learning. Our insight lies in mutually reinforcing the semantic modeling within a single item and the interaction modeling across different items. Unlike previous loosely coupled methods, we propose a novel TESPA model to deeply fuse the item semantics and interaction graph. TESPA encodes the knowledge from the user behaviors into the language models and then enriches the graph topology based on the semantics. Finally, the NLU and GNN models are co-trained under a unified learning framework in a novel multi-channel manner. TESPA achieves significant performance improvements on both on-line and off-line scenarios, demonstrating the superiority of our proposal.

## Acknowledgments

This work is supported by the National Natural Science Foundation of China (Grant No. 62372408, 62372057), and Zhejiang Key Laboratory of Accessible Perception and Intelligent Systems.

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
