# OpenReview forum: "Deeply Fusing Semantics and Interactions for Item Representation Learning via Topology-driven Pre-training"
_acmmm.org/ACMMM/2024/Conference — MM2024 Poster_

### Official Review · Reviewer_FCpY · 2024-05-26

**Rating:** 5
**Confidence:** 3

**Summary:**

This paper explores methods to enhance the joint training of GNN and NLU modules for item representation learning in recommendation scenarios. Accordingly, a new model named TESPA is proposed. TESPA begins by pre-training the NLU model using multi-grained topological signals. It then further refines the original graph topology through semantic closeness. Finally, a multi-channel training module is employed to aggregate the final item embeddings. Extensive experimental results validate the effectiveness of TESPA. Overall, the quality of this paper is good and I am positive about it.

**Strengths:**

1. The paper is well-organized and well-written.
2. This work investigates the problem of item representation learning by mutually reinforcing semantic modeling and interaction modeling.
3. The proposed TESPA consists of three parts: topology-driven pre-training of language models, semantic-based graph enrichment, and multi-channel co-training module.  Overall, the proposed solution is sound and reasonable.
4. Table 2 shows that the proposed method consistently outperforms all compared baselines, achieving significant performance improvements.
5. Comprehensive ablation experiments are provided.

**Limitations:**

1. The novelty of this paper is incremental. The overall framework of the method is based on the joint training of GNN and NLU modules for item representation learning. However, the relationship with recent similar works [15, 20, 27, 50] is not well explained, making it hard to capture the unique aspects of this paper.
2. Although this paper demonstrates the effectiveness of the designed modules through extensive experiments, it provides an insufficient explanation of the motivation behind the method. While the method's performance improvements are noteworthy, insights into why the method works should be more intriguing.
3. This paper lacks a comparative analysis between the proposed TESPA and HashCODE and Heterformer. Since these methods have similar designs (say, jointly train the GNNs and NLUs), the reasons for the performance improvements should be further elaborated.
4. It’s better to analyze the time complexity of the TESPA or efficiency performance.

**Suitability:**

3

---

### Official Review · Reviewer_GZAF · 2024-05-30

**Rating:** 5
**Confidence:** 2

**Summary:**

The work discusses the importance of learning effective item representations for e-commerce applications. It proposes a novel model called TESPA that aims to improve item representation learning by mutually reinforcing semantic modeling (capturing the intrinsic characteristics of an item) and interaction modeling (capturing relationships between items based on user behaviors like co-clicks or co-views). The key aspects of TESPA are:Encoding Interaction Signals: It encodes collaborative filtering signals from the interaction graph into language models through fine-grained topological pre-training. (i) Enriching Interaction Graph: The interaction graph is further enriched based on semantic similarities between items. (ii) Multi-Channel Co-Training: A novel multi-channel co-training paradigm is proposed to deeply fuse the semantics and interactions under a unified framework.
The authors claim that TESPA can leverage the strengths of both semantic and interaction information to facilitate better item representation learning. Experimental results from online and offline evaluations demonstrate the superiority of the proposed TESPA model.

The work is relevant and timely
The work is based two ideas: (i) to teach language models to understand interaction signals between items through continuous pre-training. (ii) exploiting the semantic similarities between the items to enhance the cross-item message passing, which addresses the data sparsity issues. Authors proposes to have a  tightly coupled co-training paradigm to deeply fuse the semantics and interactions in a multi-channel manner.

**Strengths:**

The work integrates a number of features such as to understand interaction signals between items and also exploits the semantic similarities

the work integrates graph features and NLU methods

**Limitations:**

evaluated on four datasets

ignored a  set of other metrics Recall@5,10,20, Precision@5,10,20, which are used widely used in experiments

**Suitability:**

3

---

### Official Review · Reviewer_k3Ds · 2024-06-07

**Rating:** 3
**Confidence:** 3

**Summary:**

The paper presents an approach for learning better item representations by blending semantic item information with interaction data and then training it under a unified "multi-channel co-training" framework. The author demonstrate SOTA performance on retrieval tasks.

**Strengths:**

1. The proposed approach appears to be novel, and the recipe for "Topology-Driven Pre-Training of Language Models" is an
interesting contribution.
2. Incorporation of interactions via a graph is an interesting contribution that does appear to lead to better performance.

**Limitations:**

1. The motivation for this work is fairly weak. The core contribution is "an improved item representation," but the paper
immediately delves into the technical part without formally motivating how existing representations are constructed and how
they're deficient on a technical level.

2. The figures, especially Figure 2, do not provide much information about the overall construction of the model. Furthermore, the
paper is quite dense and introduces its core contributions without formal motivation.

**Suitability:**

2

---

### Meta-Review · Area_Chair_RdZm · 2024-06-28

**Recommendation:** Accept (Poster)
**Confidence:** 4

**Metareview:**

The reviewers agree that the paper is relevant to the scope of ACM Multimedia. They are also unanimous in the opinion that it should be included in the program. Among other, they were positive about the novelty of the proposed approach and the incorporation of the interaction graph. However, they have also pointed to some drawbacks, such as a lack of in-depth motivation for the proposed approach.

Taking into account the initial reviews and the rebuttal process, I suggest acceptance of the paper in line with the reviewer recommendations.